# The Endogenous Opioid System in Schizophrenia and Treatment Resistant Schizophrenia: Increased Plasma Endomorphin 2, and κ and μ Opioid Receptors Are Associated with Interleukin-6

**DOI:** 10.3390/diagnostics10090633

**Published:** 2020-08-26

**Authors:** Shatha Rouf Moustafa, Khalid F. Al-Rawi, Drozdstoi Stoyanov, Arafat Hussein Al-Dujaili, Thitiporn Supasitthumrong, Hussein Kadhem Al-Hakeim, Michael Maes

**Affiliations:** 1Clinical Analysis Department, College of Pharmacy, Hawler Medical University, Havalan City, Erbil 44001, Iraq; shatha003@yahoo.com; 2College of Science, University of Anbar, Ramadi 31001, Iraq; sc.kfwi72@uoanbar.edu.iq; 3Department of Psychiatry, Medical University of Plovdiv, Plovdiv 4000, Bulgaria; stojanovpisevski@gmail.com; 4Clinical Psychiatry, Faculty of Medicine, University of Kufa, Najaf 540011, Iraq; arafat.aldujaili@uokufa.edu.iq; 5Department of Psychiatry, Faculty of Medicine, Chulalongkorn University, Bangkok 10110, Thailand; Thitiporn.S@chula.ac.th; 6Department of Chemistry, College of Science, University of Kufa, Najaf 540011, Iraq; headm2010@yahoo.com; 7School of Medicine, IMPACT Strategic Research Centre, Deakin University, Geelong, VIC 3220, Australia

**Keywords:** inflammation, schizophrenia, treatment resistance, neurocognition, neuroimmunomodulation

## Abstract

Background: activation of the immune-inflammatory response system (IRS) and the compensatory immune-regulatory system (CIRS) plays a key role in schizophrenia (SCZ) and treatment resistant SCZ. There are only a few data on immune and endogenous opioid system (EOS) interactions in SCZ and treatment resistant SCZ. Methods: we examined serum β-endorphin, endomorphin-2 (EM2), mu-opioid (MOR) and kappa-opioid (KOR) receptors, and interleukin (IL)-6 and IL-10 in 60 non responders to treatment (NRTT), 55 partial RTT (PRTT) and 43 normal controls. Results: serum EM2, KOR, MOR, IL-6 and IL-10 were significantly increased in SCZ as compared with controls. β-endorphin, EM2, MOR and IL-6 were significantly higher in NRTT than in PRTT. There were significant correlations between IL-6, on the one hand, and β-endorphin, EM2, KOR, and MOR, on the other, while IL-10 was significantly correlated with MOR only. A large part of the variance in negative symptoms, psychosis, hostility, excitation, mannerism, psychomotor retardation and formal thought disorders was explained by the combined effects of EM2 and MOR with or without IL-6 while increased KOR was significantly associated with all symptom dimensions. Increased MOR, KOR, EM2 and IL-6 were also associated with neurocognitive impairments including in episodic, semantic and working memory and executive functions. Conclusion: the EOS contributes to SCZ symptomatology, neurocognitive impairments and a non-response to treatment. In SCZ, EOS peptides/receptors may exert CIRS functions, whereas increased KOR levels may contribute to the pathophysiology of SCZ and EM2 and KOR to a non-response to treatment.

## 1. Introduction

The first comprehensive neuro-immune theory of schizophrenia (SCZ) was published by Smith and Maes in 1995 and suggested that activated monocytes and T-lymphocytes are key phenomena in the pathophysiology of SCZ [1]. Now, it is widely accepted that SCZ is accompanied by activation of the immune-inflammatory response system (IRS) with activated M1 macrophages as indicated by increased levels of interleukin (IL)-6, IL-1β and tumour necrosis factor-alpha (TNFα) levels, T-helper (Th)-1 cells and Th-17 cells [2,3,4,5,6,7,8]. SCZ is also accompanied by activation of the compensatory immune-regulatory system (CIRS) with increased activity of Th-2 immunocytes as indicated by increased levels of IL-4 and IL-13, and T-regulatory (Treg) cells with increased production of IL-10, and elevated levels of immune-regulatory compounds including acute phase proteins (e.g., haptoglobin) and soluble interleukin receptors such as sIL-2R, sIL-1RA, sTNFR1 and sTNFR2 [7,8]. Importantly, IRS and CIRS products including IL-1β, TNF-α, IL-6, IFN-γ, IL-4, IL-13, eotaxin (CCL11) and neurotoxic tryptophan catabolites (TRYCATs) may all cause neuroprogression and, as a consequence, deficits in memory and executive functions and, therefore, SCZ symptoms [4,6,9,10,11,12,13,14].

Treatment-resistant schizophrenia (TRS) is also accompanied by signs of IRS and CIRS including M1 (especially IL-6 trans-signaling and TNFα) and Treg (IL-10) cell activation as well as increased levels of soluble IL-2 receptor (sIL-2R), the sIL-1R antagonist (sIL-1RA), and sTNFR2 [2,4,8,15].

Endogenous opioids (EOs) are expressed in the peripheral and central nervous system where they modulate pain, reward, aversion, reinforcement, social bonding, and neurotransmitter signalling including that of glutamate and dopamine [16,17]. During IRS activation, immunocytes produce and release EOs [18] and express µ (MOR) and κ (KOR) opioid receptors [19]. In humans and animal models, EOs negatively regulate immune responses [20,21] including downregulation of cytokine and chemokine levels and associated receptors [22] and immune cell proliferation and activities [23].

Aberrations in EO system (EOS) activity are observed in SCZ, including changes in the levels of endorphins, and MOR and KOR expression [24,25]. In the post-mortem brain of SCZ patients, lowered MOR expression is observed in the striatum [25]. Because MORs mediate hedonic and social reward processing, lowered MOR expression may explain social impairments and other negative symptoms of SCZ [26]. β-Endorphin concentrations are increased in SCZ patients with negative symptoms and decreased in patients with positive symptoms as compared with controls [27]. Interestingly, clinical reports indicate that some SCZ patients are less sensitive to pain [28], suggesting increased EOS activity. Endomorphin-2, another EOS peptide, has high selectivity and affinity for MOR and mediates stress responses, sensitivity to pain, arousal, vigilance, reward, neuroendocrine and neurocognitive functions [29], suggesting that this peptide may play a role in SCZ.

In major depression, significant associations were established between increased plasma KOR/MOR levels, on the one hand, and elevated plasma IL-6 and IL-10, on the other, indicating that immune - EOS interactions play a role in the pathophysiology of depression [30]. Furthermore, IL-6 regulates the KOR gene [31] and MOR expression [32], while IL-10 increases β-endorphin gene and protein expression [33]. Since KOR/MOR/β-endorphin have immune-regulatory effects and since these products are upregulated in major depression, it was hypothesized that their increased levels might contribute to CIRS functions in depression [30]. However, in SCZ and TRS, no data are available on endomorphin 2 and possible associations between EOS compounds and IRS/CIRS functions.

Hence, this study was performed to examine serum levels of β-endorphin and endomorphin 2 as well as MOR and KOR in association with IL-6 and IL-10 in SCZ and TRS. 

## 2. Participants and Methods

### 2.1. Participants

In the current study, two groups of SCZ patients participated, namely 60 non responders to treatment (NRTT) and 55 partial responders to treatment (PRTT) and 43 healthy volunteers. The participants were of both sexes and aged 18-65 years. NRTT and PRTT were recruited at the Psychiatry Unit at Al-Imam Al-Hussain Medical City in Karbala, Iraq. All patients were diagnosed using the criteria of the diagnostic and statistical manual of mental disorders, fourth edition, *text revision* (DSM-IV-TR). NRTT was defined as non-responders to two episodes of treatment (each for at least two months), with two different antipsychotic drugs at adequate doses, without a decrease in symptom severity as screened using the Clinical Global Impression Severity (CGI-S) scale [34]. Healthy volunteers were family members or friends of staff. Patients and controls were recruited from the same catchment area, namely Karbala city, Iraq. SCZ patients who showed axis-1 DSM-IV-TR diagnoses other than SCZ were excluded from the study, including autism spectrum disorders, major depression, psycho-organic disorders, bipolar disorder, and schizoaffective disorder. Healthy controls were excluded if they had a current or lifetime diagnosis of axis I diagnosis or had a positive family history of SCZ. Moreover, patients and controls were excluded when they (a) suffered from medical disorders including chronic obstructive pulmonary disease, diabetes, rheumatoid arthritis, inflammatory bowel disease and psoriasis; and (b) suffered from neuroinflammatory or neurodegenerative disorders including Parkinson’s disease, stroke, Alzheimer’s disease and multiple sclerosis. Any subjects who had ever been taken immune-modulatory medications such as glucocorticoids were excluded from participating. We also excluded subjects who took therapeutic doses of antioxidant supplements 3 months before the blood aspiration. All participants showed serum C-reactive protein (CRP) levels <6 mg/L indicating the absence of overt inflammation.

All patients and controls, along with the guardians of patients (parents or the first-degree family members), provided written informed consent prior to participation in the study. The study was carried out according to international and Iraq ethics and privacy laws. The study was approved by the Institutional Review Board of the University of Karbala (418/ 12 July 2019), and Karbala Health Department (1331/26^th^, July, 2019), which is in agreement with the International Guidelines for Human Research protection as requested by the Declaration of Helsinki, The Belmont Report, Council for International Organizations of Medical Sciences guideline and International Conference on Harmonization in Good Clinical Practice.

### 2.2. Measurements

#### 2.2.1. Clinical Evaluations

A senior psychiatrist specialized in SCZ diagnosed the patients according to DSM-IV-TR criteria using the Mini-International Neuropsychiatric Interview (MINI), in a validated Arabic translation (Iraqi dialect). The same psychiatrist employed a semi-structured interview to assess socio-demographic and clinical data in all participants and the same day he also assessed the Clinical Global Impression (CGI) Improvement (CGI-I) and Severity (CGI-S) Scales [34]. The CGI-I was used to identify NRTT when they did not show any improvement on the CGI-I or showed poorer scores after treatment (minimally worse, much worse, very much worse). The diagnosis “responders to treatment” was made in case the CGI-I scores improved minimally, much or very much. However, since no patients showed remission after treatment, this group was named PRTT. Furthermore, we assessed the Scale for the Assessments of Negative Symptoms (SANS) to measure the severity of negative symptoms [35]. We also computed z unit-weighted composite scores to reflect the severity of psychosis, hostility, excitation, mannerism, FTD (formal thought disorders) and PMR (psychomotor retardation) [10,12,14]. Toward this end, we assessed the Brief Psychiatric Rating Scale (BPRS) [36], the Hamilton Depression Rating Scale [37] and the Positive and Negative Syndrome Scale (PANNS) [38]. The same psychiatrist also completed the Brief Assessment of Cognition in SCZ (BACS) [39] to assess episodic memory (using the List learning test); working memory (using the Digit Sequencing Task); verbal fluency (using the Controlled Word Association and Category instances tests); attention (using the symbol coding test); and executive functions (using the Tower of London). Tobacco use disorder (TUD) was examined according to DSM-IV-TR criteria. Body mass index (BMI) was calculated from the formula: body weight (kg) / squared length (m^2^). 

#### 2.2.2. Assays

Five milliliters of venous blood samples were collected after an overnight fast between 8.00 and 9.00 a.m. After clotting, blood was centrifuged at 3000 rpm for 10 min, and serum was separated and transferred into two Eppendorf tubes which were stored at -80 °C until analysis. Serum levels of IL-10 (Elabscience^®^, Inc., San Carlos, CA, USA), MOR, KOR, and endomorphin 2 (Mybiosource^®^, Inc., San Diego, CA, USA), and IL-6 and β-endorphin (Melsin Medical Co, Jilin, China) were assayed used commercial ELISA kits. All measured concentrations of β-endorphins (sensitivity = 0.1pg/mL), MOR (sensitivity = 7.18 pg/mL), KOR (sensitivity = 1.0 ng/mL), endomorphin 2 (sensitivity = 0.33 pg/mL), and IL-6 (sensitivity = 0.1 pg/mL) were greater than their assay sensitivities. There was only one IL-10 measurement (namely 4.05 pg/mL in a normal volunteer) that was lower than the sensitivity of the assay (sensitivity = 4.69 pg/mL). Therefore, we did not apply left-censoring of the data and employed the measured IL-10 concentration in our statistical analyses [4]. All intra-assay coefficients of variation were <10.0%. Serum CRP was measured using a kit supplied by Spinreact^®^, Girona, Spain, using a test based on the latex agglutination principle.

### 2.3. Statistical Analysis

We used analysis of variance or the Kruskal–Wallis test to check the differences in scale variables between categories and the Chi square test (χ^2^-test) or the ψ nominal by nominal coefficient (in cases of more than 20% of the cells having expected cell counts <5) to assess the associations between categorical variables. Associations among biomarkers, cognitive and clinical scores were computed employing Pearson’s product-moment and Spearman’s rank-order correlation coefficients. Associations between diagnosis and biomarkers were examined using multivariate general linear model (GLM) analysis while controlling for confounding variables including age, sex, TUD, BMI, and education. Consequently, we performed tests for between-subject effects to delineate the associations between diagnosis and each of the biomarkers. The effect size was assessed using partial eta-squared values (η^2^). We also computed the model-generated estimated marginal mean (SE) values and used protected pairwise comparisons among treatment means to assess differences between the diagnostic groups. Binary logistic regression analysis was employed to delineate the best predictors of SCZ versus controls and NRTT versus PRTT using the serum biomarkers levels as explanatory variables. Nagelkerke values, which were used as pseudo-R^2^ values, were computed along with Odd’s ratios with 95% confidence intervals. We used multiple regression analysis to assess the significant biomarkers, which predict the neurocognitive test results and symptom dimensions while permitting for probable effects of education, age, and sex. We used an automatic stepwise method with the inclusion of variables with a p-to-entry of 0.05 and p-to-remove of 0.06 while checking the R^2^ change. All regression analyses were checked for collinearity using tolerance and variance inflation factor values. One of the assumptions of multiple regression analysis is that the error term (between predicted and observed) is normally distributed and, therefore, we checked normality of distribution of the regression standardized residuals. Furthermore, heteroscedasticity was checked by scatterplots of residual values versus the predicted values. Tests were 2-tailed and a p-value of 0.05 was used for statistical significance. All statistical analyses were performed using IBM SPSS windows version 25, 2017. The study sample was based on power calculations using G*Power version 3.1.9.7, Heinrich-Heine-Universität Düsseldorf, Germany (downloaded freely from https://www.psychologie.hhu.de/arbeitsgruppen/allgemeine-psychologie-und-arbeitspsychologie/gpower.html). Using an effect size of 0.23, alpha = 0.05, power = 0.8 in an analysis of covariance with three groups and four covariates showed that the study sample should be *n* = 151. Using the same input parameters in a multiple regression model (with R^2^ deviation from zero) showed that a study sample of *n* = 65 would be sufficient. 

## 3. Results 

### 3.1. Socio–Demographic Data

Table 1 shows the socio–demographic data of the PRTT and NRTT and healthy controls. We included 142 SCZ patients who were treated with antipsychotic drugs during two trials with antipsychotic medications. During the first trial, patients were treated for 2 months and after this trial divided into those with a partial response (*n* = 51) and those without a clinical response (*n* = 84) (we lost seven patients through this first trial). The partial responders continued to take the similar medication for another two months while we missed again seven patients in the follow up producing a final PRTT study group of *n*= 55. The non-responders to the initial antipsychotic drug were switched to another antipsychotic drug for another 2 months and during this follow-up period we missed again thirteen patients. Two months later, eleven patients had a partial response to treatment and were categorized as PRTT, whereas sixty patients did not show any improvement on the CGI-I and, therefore, were categorized as NRTT. Consequently, fifty-five PRTT and sixty NRTT participated in the current study.

There were no significant differences in age, BMI, sex ratio, and TUD between NRTT and PRTT and normal controls. There were slightly more NRTT patients who were single as compared with the normal control group. SCZ patients were significantly more unemployed as compared with controls while years of education were slightly lower in NRTT. There were no differences in age at onset between both patients’ subgroups. All cognitive test scores (except symbol coding) were significantly different among the three study groups and the scores decreased from controls to PRTT to NRTT. We did not correct the cognitive results for type one errors due to multiple comparisons because these test results are reflective manifestations of a single underlying trait, namely a disorder in the cognitome [40]. The total SANS score was significantly different between the three study groups.

In Table 1, both the CGI-I and CGI-S scores were significantly higher in NRTT than in PRTT. The CGI-I scores in PRTT were two (much improved)–three (minimally improved) and in NRTT four (no change)–five (minimally worse). Patients in the NRTT group were more often treated with clozapine, risperidone, and quetiapine than PRTT who were more often treated with haloperidol and olanzapine.

The following symptom domains were significantly different between the three study groups and increased from controls to PRTT to NRTT: excitement (F = 320.71, df = 2/152, *p* < 0.001), FTD (F = 414.15, df = 2/152, *p* < 0.001), hostility (F = 498.12, df = 2/152, *p* < 0.001), mannerism (F = 204.41, df = 2/152, *p* < 0.001), PMR (F = 297.46, df = 2/152, *p* < 0.001) and psychosis (F = 772.55, df = 2/152, *p* < 0.001). We did not correct these data for type one errors due to multiple testing because the symptom domains are reflective manifestations of a single underlying trait, namely overall severity of schizophrenia [40]. Figure 1 shows a bar graph with the composite scores (in z scores) of these symptom domains as well as the z values of the total SANS scores.

Table 2 shows the correlations between the different biomarkers in this study (all tested at *n* = 158).

### 3.2. Differences in Biomarkers between the Study Groups

Table 3 displays the outcomes of a multivariate GLM analysis comparing the differences in the biomarkers between the three study groups while adjusting for age, BMI, and sex. There were significant differences in biomarkers between the study groups with an effect size of 0.213, while the covariates had no significant effects. Tests for between-subject effects and Table 4, which shows the estimated marginal mean (SE) values, indicate that all six biomarkers (except β-endorphin) were significantly higher in SCZ patients as compared with controls. Furthermore, β-endorphin, MOR, KOR and IL-6 were significantly higher in NRTT than in PRTT and controls. Endomorphin 2 was significantly different between the three groups and increased from controls to PRTT to NRTT. IL-10 was higher in NRTT than in controls while PRTT patients occupied an intermediate position. We also performed a multivariate GLM analysis examining the associations between the six biomarkers measured here and the diagnosis of SCZ versus normal controls while controlling for age, sex and BMI. This analysis showed that endomorphin 2, KOR, MOR, IL-6 and IL-10, but not β-endorphin (*p* = 0.865) were significantly higher is SCZ than in controls (with or without p correction for false discovery rate).

Table 5 shows the results of two binary logistic regression analyses examining the best predictors of SCZ (versus controls) and NRTT (versus PRTT) using an automatic stepwise method with biomarkers as explanatory variables while allowing for the effects of age, sex and education. The first regression analysis showed that SCZ was best explained by increased levels of endomorphin 2, KOR, and IL-10 (χ^2^ = 39.338, df = 3, *p* < 0.001, Nagelkerke = 0.319) with an accuracy of 72.2%, sensitivity of 74.8% and specificity of 65.1%. The second regression shows that the combination of endomorphin 2, MOR, and IL-6 best discriminated NRTT from PRTT (χ^2^ = 34.47, df = 3, *p* < 0.001, Nagelkerke = 0.346) with an accuracy of 72.2%, sensitivity of 80.0% and a specificity of 76.5%.

### 3.3. Effects of Background Variables

As shown above, age, sex, and BMI had no significant effects on serum biomarker levels. TUD also had no significant effect on the measured biomarker levels (F = 0.19, df = 6/146, *p* = 0.979, partial η^2^ = 0.008). We also examined the effects of antipsychotic drug administration using multivariate GLM analysis and tests for between-subjects. We found no significant effects of use of clozapine (F = 0.63, df = 6/146, *p* = 0.704, partial η^2^ = 0.025), haloperidol (F = 0.88, df = 6/146, *p* = 0.513, Partial η^2^ = 0.035), risperidone (F = 0.92, df = 6/146, *p* = 0.484, partial η^2^ = 0.036), and quetiapine (F = 0.93, df = 6/146, *p* = 0.476, partial η^2^ = 0.037). However, use of olanzapine had a significant effect on the measured biomarkers (F = 2.89, df = 6/146, *p* = 0.011, partial η^2^ = 0.106), although after p-correction for multiple testing this effect was no longer significant (*p* = 0.195).

### 3.4. Prediction of Symptom Domains by Biomarkers

Table 6 shows different stepwise multiple regression analyses with the symptom domains as dependent variables and the six biomarkers as explanatory variables while allowing for the effects of education, age and sex. Regression #1 shows that 28.0% of the variance in the total SANS score was explained by the regression on endomorphin 2, MOR, and IL-6. Regressions #2, #3, #4 and #5 show that the same variables explained a considerable part of the variance in psychosis (28.3%, but additionally with education), hostility (26.9%), excitation (22.4%), and PMR (28.4%). Regression #6 shows that 23.7% of the variance in mannerism was explained by endomorphin 2, MOR, KOR, and IL-10. Regression #7 shows that 27.8% of the variance in FTD was explained by the combined effects of endomorphin 2, MOR, IL-6, IL-10, and education. Partial correlation coefficients (adjusted for age, sex, and education) showed that all symptom profiles were significantly associated with plasma KOR kevels (all r > 0.314, *p* < 0.001, *n* = 153), but not with β-endorphin.

### 3.5. Prediction of Cognitive Impairments by Biomarkers

Table 7 shows the outcome of six multiple regression analyses with the cognitive test results as dependent variables and biomarkers as explanatory variables while allowing for the effects of age, sex and education. We found that (regression #1) 16.3% of the variance in List Learning scores was explained by the regression on IL-6 and MOR (all inversely associated). Up to 21.4% of the variance in Digit Sequencing Task scores (regression #2) was explained by the combined effects of MOR, IL-6, endomorphin 2 (inversely) and education (positively). Part of the variance (20.2%) in Category Instances test scores (regression #3) was explained by MOR, endomorphin 2 (negatively) and education (positively). We found that 28.3% of the variance in the Controlled Oral Word Association Test (COWA) test scores (regression #4) was explained by the cumulative effects of MOR, IL-6, endomorphin 2, and age (negatively) while 22.9% of the variance in symbol coding scores (#5) was negatively associated with KOR, IL-10, and endomorphin 2. Up to 29.3% of the variance in Tower of London test scores (regression #6) was explained by the combined effects of MOR, IL-6, endomorphin 2, sex (all inversely) and education (positively). Partial correlation coefficients (adjusted for age, sex, and education) showed that all cognitive tests results were significantly associated with plasma KOR (all r > 0.206, *p* < 0.01, *n* = 152), but not with β-endorphin.

## 4. Discussion

The first major finding of this study is that serum levels of KOR, MOR, endomorphin 2, IL-6 and IL-10 are significantly increased in SCZ as compared with controls. This is a first report on increased endomorphin 2 levels in SCZ. Volk et al. found that increased MOR mRNA and protein levels in SCZ are largely independent of illness severity, suggesting that increased MOR expression is part of the disease process rather than a consequence of illness chronicity [41]. In Han Chinese, a MOR polymorphism may confer risk for SCZ [42] while an A118G polymorphism of the MOR gene was associated with SCZ [43]. One study reported lowered MOR availability in the brain of SCZ patients who died as a result of suicide, which would be consistent with increased levels of EOS peptides occupying those receptors [44]. Our negative findings on β-endorphin levels in SCZ are not in agreement with those of a previous report [42]. Animal models of SCZ are accompanied by moderate alterations in EOS peptides [24,25,28,41]. Our findings on increased IL-6 and IL-10 levels in SCZ are in agreement with previous reports that SCZ is accompanied by enhanced IRS and CIRS functions [8]. 

This is also a first report that a non-response to treatment is characterized by increased MOR, endomorphin 2 and β-endorphin levels, indicating that the EOS participates in the pathophysiology of treatment resistance. Our findings that IL-6 is increased in NRTT as compared with PRTT is in accordance with previous reports that IL-6 and macrophage M1 activation are associated with TRS [8,15]. In the current study we found that serum IL-10 was significantly higher in NRTT than in controls, whereas PRTT occupied an intermediate position. Previous reports showed that TRS is accompanied by increased IL-10 levels [45].

The second major finding of this study is that IL-6 levels are strongly associated with all four EOS biomarkers whilst IL-10 is associated with MOR concentrations only. Similar findings were reported in major depression showing significant associations between an immune activation index (based on IL-6 and IL-10) and KOR, MOR and β-endorphin levels [5,30]. Many inflammatory diseases are associated with up-regulation of opioid receptors [46] and, in those conditions, inflammation is associated with increased MOR sensitivity in the periphery and in the brainstem [47]. Moreover, the combination of IL-6 with endomorphin 2 and MOR yielded a highly significant discrimination of NRTT from PRTT indicating that upregulation of the immune-EOS axis is associated with the pathophysiology of a non-response to treatment.

During activation of the IRS, immune cells produce a) opioid peptides, [48,49] with increased EO concentrations in blood and inflammatory sites [50]. MOR and KOR are widely distributed on peripheral blood immune cells [51,52,53,54,55,56] while the expression of the MOR gene in immune and neuronal cells is enhanced by pro-inflammatory signals including IL-1β, IL-6 and TNF-α [57]. Moreover, LPS, which is increased in deficit schizophrenia [58], increases MOR transcription in macrophages via oxidative stress signals [59]. As such, proliferation and activation of immune cells may be accompanied with increased expression of opioid receptors while the toxic effects of inflammatory and oxidative stress mechanisms may damage cell membranes thereby increasing shedding of the opioid receptors into the plasma. Some T-cell subsets release cytokines and EOS peptides/receptors that can promote, suppress, or resolve pain [60,61]. Moreover, also endomorphin 2 may be produced by immunocytes in inflamed subcutaneous tissues, whereas this peptide is almost absent in non-inflamed tissue [62]. In addition, MOR is expressed in macrophages and neutrophils, indicating that endomorphins produced during inflammation can stimulate MORs on the surface of these cells [63]. As such, the positive intercorrelations detected in our study between increased IL-6 levels and EOS biomarkers, including endomorphin 2, suggest that the enhanced IRS/CIRS responses in SCZ patients and especially in NRTT are accompanied by increased production of EOS biomarkers. This may be important to brain functions as some EOS peptides including endomorphin 2 may cross the blood–brain barrier (BBB) following intraperitoneal administration [64]. Ting et al. (1997) reported that increased MOR/KOR binding on the BBB precedes BBB disruption [65]. Moreover, recent research shows that SCZ and especially deficit SCZ is accompanied by a breakdown of the tight junction and vascular barriers of the BBB [66].

There is now evidence that the EOS exerts immune-regulatory activities through multiple feedback loops [67,68]. For example, EOS peptides/receptors modulate adaptive immune functions including attenuating Th functions and neutrophil chemotaxis [69,70], increasing T cell apoptosis [70] and levels of immune-regulatory cytokines including IL-10 [71]. Increased MOR, KOR and β-endorphin levels display negative immune-regulatory properties as reviewed in [68]. Therefore, the latter authors concluded that, in major depression, increased dynorphin/KOR and β-endorphin/MOR signaling may contribute to CIRS functions [68]. In addition, endomorphin 2 may regulate neutrophil, macrophage and microglia functions even at ultra-low concentrations [72]. Yang et al. (2012) reported that activated dendritic cells induce their expression and secretion of endomorphins and that the latter, in turn, may suppress T lymphocyte proliferation through stimulation of MOR [73]. Endomorphin 2 not only regulates the production of pro-inflammatory cytokines, but also inhibits macrophage chemotaxis and the production of reactive oxygen species (ROS) by macrophages and neutrophils [72,74]. Importantly, in other inflammatory disorders, endomorphins exert anti-inflammatory actions [75,76] for example by inhibiting IL-8, but not IL-6 [76]. Moreover, endomorphin 2 is advocated as an agent to treat chronic inflammatory disease [75]. As such, all four EOS biomarkers measured here are produced by activated immunocytes and may, consequently, exert CIRS activities thereby regulating the immune response in SCZ patients and NRTT.

The third major finding of this study is that MOR, KOR and endomorphin 2 were associated with PHEMN (psychosis, hostility, excitation, mannerism, negative) symptoms, psychomotor retardation and formal thought disorders as well as neurocognitive deficits in memory, attention and executive functions. The detrimental effects of IL-6 and, especially, IL-6 trans-signaling on brain functions including neurocognitive functions are well established [77,78,79]. Although the EOS may exert CIRS (see above discussion) and neuroprotective effects [73], this system may have detrimental effects as well. For example, endomorphins may activate immune pathways, which may result in more detrimental effects, including increased IL-1β signaling, macrophage adhesion, expression of adhesion molecules on macrophages, and neutrophil chemotaxis [72,74].

Moreover, KOR agonists may exhibit psychotomimic properties while opioid antagonists may ameliorate SCZ symptoms [80]. KOR activation and administration of KOR agonists including salvinorin A may induce hallucinations, anxiety, depression, negative-like symptoms (lack of motivation and social withdrawal), psychomotor retardation, dysphoria and neurocognitive impairments including in attention, working memory and task performance, which are quite similar to the effects of acute ketamine administration [81,82,83]. These effects were explained by attenuated glutamate and dopamine release in the prefrontal cortex, which play a key role in neuropsychological functioning [84,85]. Therefore, increased KOR expression as observed in our study may play a role in the pathophysiology of SCZ.

Moreover, endomorphin 2 may stimulate postsynaptic MORs causing postsynaptic hyperpolarization of excitatory interneurons [86,87]. Endomorphin 2 may additionally induce excitation, a bell-shaped dose–response curve for locomotor enhancement and aversive effects, and place aversion [88]. As such, increased endomorphin 2 levels could, combined with increased KOR, contribute to the pathophysiology of a non-response to treatment.

This study should be interpreted with regard to its limitations. First, this is a case control study and, therefore, no firm conclusions can be drawn on causal relationships. Second, it would have been even more interesting if we had measured other IRS and CIRS cytokines and endomorphin 1. Nevertheless, here we assayed endomorphin 2 as it may have additional functions with regard to the pathophysiology of schizophrenia as compared with endomorphin 1 including effects on the release of dynorphin A and [Met]enkephalin, and the consequent activation of KOR- and delta-opioid receptors [89,90]. Moreover, dynorphin released in response to endomorphin 2 has a major role in immune cell functions [91]. Third, the genes encoding endomorphin 1 and 2 have not yet been identified [92], although it is suggested that endomorphins may be synthesized enzymatically, by a non-ribosomal mechanism [93]. Fourth, it is plausible that mass spectrometry may be more suitable and reliable to detect endomorphin 2 levels as compared with ELISA measurements, although enzyme immunoassay kits may detect low endomorphin 2 levels in rodent plasma and cerebro-spinal fluid [94]. 

## 5. Conclusions

Serum levels of EOS biomarkers including endomorphin-2, MOR, KOR, IL-6 and IL-10 are increased in SCZ patients as compared with controls, while increased endomorphin 2, MOR, and IL-6 are biomarker features of a non-response to treatment. The findings indicate that changes in the EOS and immune–EOS interactions play a role in the pathophysiology of SCZ and a non-response to treatment.

## Figures and Tables

**Figure 1 diagnostics-10-00633-f001:**
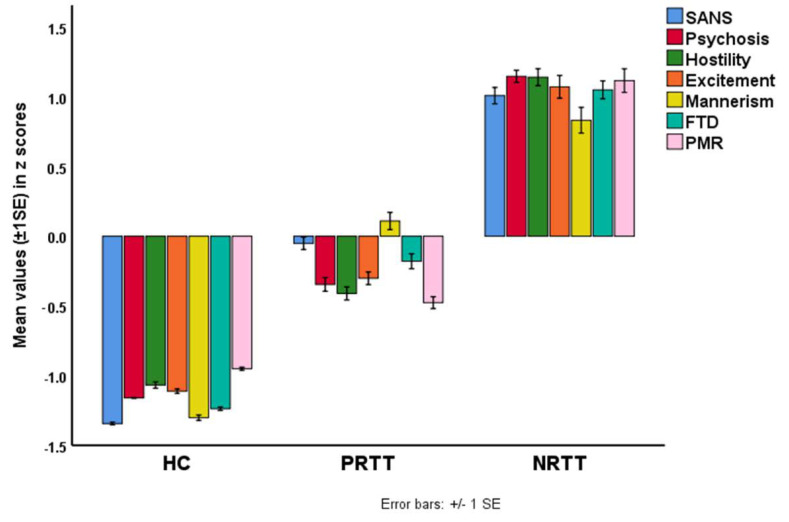
Bar plot showing the mean (±1 SE) z scores of psychosis, hostility, excitement, mannerism, formal thought disorders (FTD), psychomotor retardation (PMR) and the total score on the Scale for the Assessment of Negative Symptoms (SANS) in healthy controls (HC), partial responders to treatment (PRTT) and non-responders to treatment (NRTT).

**Table 1 diagnostics-10-00633-t001:** Demographic and clinical data of healthy controls (HC) and partial (PRTT) and non (NRTT) responders to treatment.

Variables	HC ^A^ (*n* = 43)	PRTT ^B^ (*n* = 55)	NRTT ^C^ (*n* = 60)	F/φ/χ^2^	df	*p*
Age (years)	33.2 (11.1)	36.5 (9.5)	36.2 (12.3)	F = 1.29	2/155	0.280
Sex (Female/Male)	19/24	15/40	22/38	χ^2^ = 3.08	2	0.214
Married (No/Yes)	12/31 ^C^	35/30	32/28 ^A^	χ^2^ = 6.69	2	0.035
BMI (kg/m^2^)	27.9 (4.1)	29.6 (4.3)	28.4 (4.9)	F = 1.90	2/155	0.153
TUD (No/Yes)	30/13	44/11	40/20	χ^2^ = 2.71	2	0.258
Employment (No/Yes)	17/26 ^B,C^	36/19 ^A^	43/17 ^A^	χ^2^ = 11.63	2	0.003
Education (years)	11.1 (3.6) ^C^	10.8 (4.5) ^C^	8.9 (4.7) ^A,B^	F = 4.21	2/155	0.017
Age at onset (years)		27.5 (7.5)	29.3 (10.2)	F = 1.14	1/113	0.287
List learning *	0.797 (0.596)	0.437 (0.580)	−0.959 (0.688)	KW		<0.001
Digit sequencing task *	1.369 (0.530)	−0.221 (0.604)	−0.755 (0.322)	KW		<0.001
Category instances *	0.821 (0.398)	0.136 (0.641)	−0.693 (1.064)	KW		<0.001
COWA *	1.386 (0.373)	−0.149 (0.515)	−0.833 (0.399)	KW		<0.001
Symbol coding	1.559 (0.403) ^B,C^	−0.518 (0.216) ^A^	−0.617 (0.267) ^A^	KW		<0.001
Tower of London *	1.195 (0.463)	0.131 (0.709)	−0.848 (0.486)	KW		<0.001
SANS total score *	11.2 (5.0)	52.6 (12.2)	91.9 (17.0)	KW		<0.001
CGI-I		2.73 (0.45)	4.20 (0.40)	F = 342.92	1/113	<0.001
CGI-S		4.38 (0.49)	5.95 (0.70)	F = 190.63	1/113	<0.001
Clozapine (No/Yes)		55/0	46/14	φ = 0.356		<0.001
Quietiapin (No/Yes)		55/0	54/6	φ = 0.225		0.016
Haloperidol (No/Yes)		43/12	60/0	φ = 0.357		<0.001
Olanzapine (No/Yes)		2/53	25/35	φ = 0.448		<0.001
Risperidone		53/2	48/12	φ = 0.250		0.007

Results are shown as mean (SD), except the neuropsychological test scores which are shown as mean (SD) values after considering the effects of age, sex and years of education. All results of general linear model (GLM) analysis (F), analysis of contingency tables (χ^2^), nominal x nominal coefficient φ, or the Kruskal–Wallis (KW) test. ^A,B,C^: results of pairwise comparisons of ratios or mean values between the diagnostic groups; the letters indicate which columns are significantly different, for example ^c^ indicates different from group c that is non responders to treatment (NRTT). * These test scores are significantly different between the three study groups. BMI: Body mass Index; COWA: Controlled Oral Word Association Test; CGI-I: Clinical Global Impression-Improvement scale; CGI-S: Clinical Global Impression- Severity scale; SANS: Scale for the Assessment of Negative Symptoms; TUD: Tobacco use disorder.

**Table 2 diagnostics-10-00633-t002:** Correlation matrix between the biomarkers included in this study.

Variables	IL-6	IL-10	β-Endorphin	Endomorphin 2	KOR	MOR
IL-6						
IL-10	0.152					
β-endorphin	0.173 *	0.090				
Endomorphin 2	0.253 **	0.005	0.254 **			
KOR	0.491 **	0.084	0.113	0.385 **		
MOR	0.333 **	0.289 **	0.090	0.015	0.449 **	

* *p* < 0.05, ** *p* ≤ 0.001, IL: interleukin; KOR: κ-opioid receptor; MOR: µ-opioid receptor.

**Table 3 diagnostics-10-00633-t003:** Results of multivariate GLM analysis showing the associations between biomarkers and diagnosis while adjusting for background variables.

Type	Dependent Variables	Explanatory Variables	F	df	*p*	Partial η^2^
Multivariate	β-Endorphin, Endomorphin 2, KOR, MOR, IL-6, IL-10	Diagnosis	6.68	12/296	<0.001	0.213
		Sex	1.23	6/147	0.296	0.048
		Age	1.14	6/147	0.341	0.045
		BMI	0.68	6/147	0.667	0.027
Tests for between-subject effects	β-Endorphin	Diagnosis	4.25	2/152	0.016	0.053
	Endomorphin 2	Diagnosis	13.44	2/152	<0.001	0.150
	KOR	Diagnosis	13.38	2/152	<0.001	0.150
	MOR	Diagnosis	14.71	2/152	<0.001	0.162
	IL-6	Diagnosis	15.22	2/152	<0.001	0.167
	IL-10	Diagnosis	3.56	2/152	0.031	0.045

Diagnosis: partial responders to treatment versus non-responders to treatment versus healthy controls. BMI: body mass index; IL: interleukin; KOR: κ-opioid receptor; MOR: µ-opioid receptor.

**Table 4 diagnostics-10-00633-t004:** Model-generated estimated marginal means values (SE) of the biomarkers in partial responders to treatment (PRTT), non-responders to treatment (NRTT) and healthy controls (HC).

Biomarkers	HC ^A^	PRTT ^B^	NRTT ^C^
β-Endorphin (pg/mL)	20.37 (2.52)	16.57 (2.32) ^C^	24.62 (2.14) ^B^
Endomorphin 2 (pg/mL)	256.84 (39.69) ^B,C^	315.77 (36.61) ^A,C^	478.08 (33.71) ^A,B^
KOR (ng/mL)	4.24 (1.07) ^B,C^	7.70 (0.98) ^A^	7.32 (0.91) ^A^
MOR (pg/mL)	3.03 (0.36) ^C^	3.59 (0.34) ^C^	4.85 (0.31) ^A,B^
IL-6 (pg/mL)	4.82 (0.86) ^C^	5.73 (0.80) ^C^	7.79 (0.73) ^A,B^
IL-10 (pg/mL)	10.83 (0.87) ^C^	12.59 (0.80)	14.12 (0.74) ^A^

^A,B,C^: pairwise comparisons between group means. IL: interleukin; KOR: κ-opioid receptor; MOR: µopioid receptor.

**Table 5 diagnostics-10-00633-t005:** Results of two different binary logistic regression analyses with schizophrenia (versus healthy controls) and non-responders to treatment (NRTT) versus partial responders to treatment (PRTT) as dependent variables and the biomarkers as explanatory variables.

Dichotomies	Explanatory Variables	B	SE	Wald	df	*p*	OR	95% CI
**Schizophrenia/Controls**	Endomorphin 2	0.496	0.237	4.386	1	0.036	1.642	1.032–2.61
	KOR	0.979	0.280	12.231	1	<0.001	2.663	1.538–4.61
	IL-10	0.591	0.225	6.902	1	0.009	1.806	1.162–2.81
**NRTT/PRTT**	Endomorphin 2	0.711	0.261	7.434	1	0.006	2.037	1.22–3.40
	MOR	0.673	0.260	6.705	1	0.010	1.960	1.18–3.26
	IL-6	0.757	0.258	8.600	1	0.003	2.132	1.29–3.54

B: logistic regression coefficient, OR: Odds ratio, 95% CI: 95% confidence intervals. IL: interleukin; KOR: κ-opioid receptor; MOR: µ-opioid receptor.

**Table 6 diagnostics-10-00633-t006:** Results of multiple regression analysis with schizophrenia symptom domains as dependent variables.

Dependent Variables	Explanatory Variables	β	t	*p*	F _model_	df	*p*	R^2^
**#1. SANS**	**Model**				19.94	3/154	<0.001	0.280
	Endomorphin 2	0.302	4.290	<0.001				
	MOR	0.268	3.724	<0.001				
	IL-6	0.191	2.600	0.010				
**#2. Psychosis**	**Model**				15.13	4/153	<0.001	0.283
	Endomorphin 2	0.259	3.674	<0.001				
	MOR	0.263	3.644	<0.001				
	IL-6	0.188	2.511	0.013				
	Education	–0.157	–2.248	0.026				
**#3. Hostility**	**Model**				18.91	3/154	<0.001	0.269
	Endomorphin 2	0.246	3.460	0.001				
	MOR	0.270	3.724	<0.001				
	IL-6	0.231	3.128	0.002				
**#4. Excitation**	**Model**				14.82	3/154	<0.001	0.224
	Endomorphin 2	0.218	2.984	0.003				
	MOR	0.248	3.314	0.001				
	IL-6	0.215	2.822	0.005				
**#5. PMR**	**Model**				20.39	3/154	<0.001	0.284
	Endomorphin 2	0.310	4.412	<0.001				
	MOR	0.223	3.104	0.002				
	IL-6	0.233	3.189	0.002				
**#6. Mannerism**	**Model**				11.91	4/153	<0.001	0.237
	Endomorphin 2	0.170	2.252	0.026				
	MOR	0.183	2.310	0.022				
	KOR	0.211	2.613	0.010				
	IL-10	0.199	2.710	0.008				
**#7. FTD**	**Model**				11.71	5/152	<0.001	0.278
	Endomorphin 2	0.203	2.850	0.005				
	MOR	0.246	3.285	0.001				
	IL-6	0.168	2.223	0.028				
	Education	–0.159	–2.254	0.026				
	IL-10	0.147	2.047	0.042				

β: standardized regression coefficient, IL: interleukin; KOR: κ-opioid receptor; MOR: µ-opioid receptor. SANS: Scale for the Assessment of Negative Symptoms; FTD: formal thought disorders; PMR: psychomotor retardation, t: t-statistic value, F: F-statistic value, df: degree of freedom, and #: number of regression analysis.

**Table 7 diagnostics-10-00633-t007:** Results of multiple regression analysis with neurocognitive test scores as dependent variables.

Dependent Variables	Explanatory Variables	β	t	*p*	F _model_	df	*p*	R^2^
**#1. List learning**	**Model**				15.08	2/155	<0.001	0.163
	MOR	0.304	3.945	<0.001				
	IL-6	0.188	2.441	0.016				
**#2. Digit sequencing task**	**Model**				10.43	4/153	<0.001	0.214
	MOR	0.224	2.965	0.004				
	IL-6	0.190	2.431	0.016				
	Endomorphin 2	0.158	2.143	0.034				
	Education	0.188	2.575	0.011				
**#3. Category instances**	**Model**				13.03	3/154	<0.001	0.202
	MOR	0.283	3.891	<0.001				
	Endomorphin 2	0.246	3.379	0.001				
	Education	0.192	2.672	0.008				
**#4. COWA**	**Model**				15.12	4/153	<0.001	0.283
	MOR	0.293	4.070	<0.001				
	IL-6	0.183	2.497	0.014				
	Endomorphin 2	0.259	3.669	<0.001				
	Age	0.161	2.376	0.019				
**#5. Symbol coding**	**Model**				15.15	3/153	<0.001	0.229
	KOR	0.306	4.013	<0.001				
	IL-10	0.201	2.826	0.005				
	Endomorphin 2	0.201	2.647	0.009				
**#6. Tower of London**	**Model**				12.61	5/152	<0.001	0.293
	MOR	0.273	3.783	<0.001				
	IL-6	0.150	2.007	0.046				
	Endomorphin 2	0.146	2.070	0.040				
	Sex	0.183	2.656	0.009				
	Education	0.290	4.178	<0.001				

COWA: Controlled Oral Word Association Test; IL: interleukin; KOR: κ-opioid receptor; MOR: µ-opioid receptor. FTD: formal thought disorders; PMR: psychomotor retardation; SANS: Scale for the assessment of negative symptoms, β: standardized regression coefficient with t-statistic and p-value, F: F-statistic value, df: degree of freedom, and #: number of regression analysis.

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
