# Peer review of "The Endogenous Opioid System in Schizophrenia and Treatment Resistant Schizophrenia: Increased Plasma Endomorphin 2, and κ and μ Opioid Receptors Are Associated with Interleukin-6"

_diagnostics, 2020, doi:10.3390/diagnostics10090633_

Round 1

Reviewer 1 Report

The paper:  “The endogenous opioid system in schizophrenia and treatment resistant schizophrenia: increased plasma endomorphin 2, and κ and μ opioid receptors are associated with interleukin-6” written by Shatha Moustafa, Khalid F Al-Rawi, Drozdstoi Stoyanov, Arafat Hussein Al-Dujaili, Thitiporn Supasitthumrong, Hussein Al-Hakeim, Michael Maes is very interesting, important, and original enough to be published in Diagnostitics and deserves to be fast-tracked for prompt publication. Data analyses were conducted in very detail with outstanding expertise in statistical methods. Recapitulation of the obtained results in the Discussion demonstrates  contribution of endogenous opioid system to schizophrenia symptomatology, neurocognitive impairments and a non-response to treatment.  Significance of the presented observations and unique knowledge of authors of the role of the immune-inflammatory system (IRS), the compensatory immune-regulatory system (CIRS) and  endogenous opioid system in schizophrenia are worth to admit.

It is necessary to add a list of used abbreviations with their explanation.

On the title page of the publication, when specifying the place of employment of the authors, it is not explained in which country the University of Anbar and Deakin University are located.

In the Abstract, the abbreviation SCZ is not explained.

In the second sentence of Introduction, the word "levels" is used twice and this sentence should be corrected.

IL-10 produces a lot of lymphocyte subtypes, not just Treg lymphocytes, and this information should be included in the Introduction.

The abbreviations sIL-2R, sIL-RA used for the first time in verse 49 are only explained in verse 57.

The way in which opioid receptor names are written should be harmonized (verse: 20, 62, 63,……).

Clear, understandable explanations of which patients were included in the NRTT and PRTT groups should already be included in the Introduction.

Author Response

The paper:  “The endogenous opioid system in schizophrenia and treatment resistant schizophrenia: increased plasma endomorphin 2, and κ and μ opioid receptors are associated with interleukin-6” written by Shatha Moustafa, Khalid F Al-Rawi, Drozdstoi Stoyanov, Arafat Hussein Al-Dujaili, Thitiporn Supasitthumrong, Hussein Al-Hakeim, Michael Maes is very interesting, important, and original enough to be published in Diagnostitics and deserves to be fast-tracked for prompt publication. Data analyses were conducted in very detail with outstanding expertise in statistical methods. Recapitulation of the obtained results in the Discussion demonstrates  contribution of endogenous opioid system to schizophrenia symptomatology, neurocognitive impairments and a non-response to treatment.  Significance of the presented observations and unique knowledge of authors of the role of the immune-inflammatory system (IRS), the compensatory immune-regulatory system (CIRS) and  endogenous opioid system in schizophrenia are worth to admit.

It is necessary to add a list of used abbreviations with their explanation.

@ANSWER: ADDEDD see lines 38-48.

On the title page of the publication, when specifying the place of employment of the authors, it is not explained in which country the University of Anbar and Deakin University are located.

@ANSWER: added, see lines 8-14

In the Abstract, the abbreviation SCZ is not explained.

@ANSWER: added: see line 17  

In the second sentence of Introduction, the word "levels" is used twice and this sentence should be corrected.

@ANSWER: deleted, see line 55

IL-10 produces a lot of lymphocyte subtypes, not just Treg lymphocytes, and this information should be included in the Introduction.

@ANSWER: added: see line 59-60

The abbreviations sIL-2R, sIL-RA used for the first time in verse 49 are only explained in verse 57.

@ANSWER: corrected, see lines 61-62

The way in which opioid receptor names are written should be harmonized (verse: 20, 62, 63,……).

@ANSWER: corrected, see lines 79-80

Clear, understandable explanations of which patients were included in the NRTT and PRTT groups should already be included in the Introduction.

@ANSWER: added, see lines 71-74

Reviewer 2 Report

In the manuscript of SR Moustafa et al. blood serum levels of endogenous opioid and inflammatory biomarkers such as endomorphin-2, MOR, KOR, IL-6 and IL-10 are increased in schizophrenic patients as compared with control individuals, while increased endomorphin 2, MOR, and IL-6 levels are observed as biomarker features of people whose non-responded to antipsychotic treatments.  The reported experimental observations may be of interest in understanding the disease, possibly also in the treatment of schizophrenia, however, the reviewer found several points of doubt in the report to which the authors should respond.

Are the number of subjects in each group really sufficient for the multivariate statistical analyzes presented? 

The formation of beta-endorphin is well known (cleavage product of the precursor POMC), unfortunately the biosynthesis of the two types of endomorphin tetrapeptide amides (endomorphin1 and 2) is not at all clear, so I think this makes no sense for diagnosis. Why endomorphin 1 was not studied?

Both MOR and KOR are cell surface, strongly membrane integrated receptor proteins. By what assumption do the authors think that these proteins may be present in the blood serum, and even a diagnosis can be made based on changes in their amount?

I consider mass spectrometry to be more suitable and reliable for the detection of endomorphin 2 compared to Elisa measurements. 

How do the authors explain the three order of magnitude quantitative differences observed between MOR (pg/ml) and KOR (ng/ml) levels?

Author Response

In the manuscript of SR Moustafa et al. blood serum levels of endogenous opioid and inflammatory biomarkers such as endomorphin-2, MOR, KOR, IL-6 and IL-10 are increased in schizophrenic patients as compared with control individuals, while increased endomorphin 2, MOR, and IL-6 levels are observed as biomarker features of people whose non-responded to antipsychotic treatments.  The reported experimental observations may be of interest in understanding the disease, possibly also in the treatment of schizophrenia, however, the reviewer found several points of doubt in the report to which the authors should respond.

Are the number of subjects in each group really sufficient for the multivariate statistical analyzes presented? 

@ANSWER: number of patients was based on power calculation. See statistics, lines 195-199.

The formation of beta-endorphin is well known (cleavage product of the precursor POMC), unfortunately the biosynthesis of the two types of endomorphin tetrapeptide amides (endomorphin1 and 2) is not at all clear, so I think this makes no sense for diagnosis. Why endomorphin 1 was not studied?

@ANSWER: this is now addressed in the limitation section, see lines 440-443.

It reads:  Third, the genes encoding endomorphin 1 and 2 have not yet been identified (Offermanns and Rosenthal 2008), although it is suggested that endomorphins may be synthesized enzymatically, by a non-ribosomal mechanism (Terskiy et al. 2007). In addition, in this study we did not use the EOS markers as diagnostic criteria. If so, we would have use cross-validation techniques in SVM or neural networks. All figures of merit presented here are used as effect sizes (including diagnostic performance).

And lines: 434-440. It reads: 

Second, it would have been even more interesting if we had measured other IRS and CIRS cytokines and endomorphin 1. Nevertheless, here we assayed endomorphin 2 as it may have additional functions with regard to the pathophysiology of schizophrenia as compared with endomorphin 1 including effects on the release of dynorphin A and [Met]enkephalin, and the consequent activation of KOR- and delta-opioid receptors (Willis and Coggeshall, 2012; Sakurada et al., 2008). Moreover, dynorphin released in response to endomorphin 2 has a major role in immune cell functions (Gein 2014).

Both MOR and KOR are cell surface, strongly membrane integrated receptor proteins. By what assumption do the authors think that these proteins may be present in the blood serum, and even a diagnosis can be made based on changes in their amount?

@ANSWER: addressed lines 364-372. It reads:

MOR and KOR are widely distributed on peripheral blood immune cells (Bidlack, 2000; Suzuki et al., 2001; 2002; Rafaelli et al., 2020; Machelska and Celik, 2020; Maher et al., 2020; Scharp, 2006) while the expression of the MOR gene in immune and neuronal cells is enhanced by pro-inflammatory signals including IL-1β, IL-6 and TNF-α (Kraus, 2009). Moreover, LPS, which is increased in deficit schizophrenia (Maes et al., 2019a), increases MOR transcription in macrophages via oxidative stress signals (Langsdorf et al., 2011). As such, proliferation and activation of immune cells may be accompanied with increased expression of opioid receptors while the toxic effects of inflammatory and oxidative stress mechanisms may damage cell membranes thereby increasing shedding of the opioid receptors into the plasma.

I consider mass spectrometry to be more suitable and reliable for the detection of endomorphin 2 compared to Elisa measurements.

@ANSWER: addressed in the text, see lines 443-446. Four, it is plausible that mass spectrometry may be more suitable and reliable to detect endomorphin 2 levels as compared with Elisa measurements, although enzyme immunoassay kits may detect low endomorphin 2 levels in rodent plasma and cerebro-spinal fluid (Kou et al., 2009). 

How do the authors explain the three order of magnitude quantitative differences observed between MOR (pg/ml) and KOR (ng/ml) levels?

@ANSWER: Some substances are more expressed than others. I think it is like sodium blood concentrations which are higher than blood potassium levels.

Round 2

Reviewer 2 Report

I have accepted the changes made by the authors and suggest the publication of the revised version.